# The Effect of Breaking Up Sedentary Time with Calisthenics on Neuromuscular Function: A Preliminary Study

**DOI:** 10.3390/ijerph192114597

**Published:** 2022-11-07

**Authors:** Emily Mear, Valerie Frances Gladwell, Jamie Pethick

**Affiliations:** 1School of Sport, Rehabilitation and Exercise Sciences, University of Essex, Colchester CO4 3SQ, UK; 2Institute of Health and Wellbeing, University of Suffolk, Ipswich IP4 1QJ, UK

**Keywords:** sedentary behaviour, breaking up sedentary time, calisthenics, neuromuscular, strength, force steadiness, balance

## Abstract

The ageing process results in reduced neuromuscular function. This alongside prolonged sedentary behaviour is associated with decreased muscle strength, force control and ability to maintain balance. Breaking up sedentary time with regular bouts of physical activity has numerous health benefits, though the effects on neuromuscular function are unknown. This study investigated the effect of breaking up sedentary time with calisthenic exercise on neuromuscular function. 17 healthy adults (33 ± 13.1 years), who spent ≥6 h/day sitting, were assigned to a four-week calisthenics intervention (*n* = 8) or control group (*n* = 9). The calisthenics intervention involved performing up to eight sets of exercises during the working day (09:00–17:00); with one set consisting of eight repetitions of five difference exercises (including squats and lunges). Before and immediately after the intervention, measures of knee extensor maximal voluntary contraction (MVC) and submaximal force control (measures of the magnitude and complexity of force fluctuations), and dynamic balance (Y balance test) were taken. The calisthenics intervention resulted in a significant increase in knee extensor MVC (*p* = 0.036), significant decreases in the standard deviation (*p* = 0.031) and coefficient of variation (*p* = 0.016) of knee extensor force fluctuations during contractions at 40% MVC, and a significant increase in Y balance test posterolateral reach with left leg stance (*p* = 0.046). These results suggest that breaking up sedentary time with calisthenics may be effective at increasing muscle strength, force steadiness and dynamic balance all of which might help reduce the effects of the ageing process.

## 1. Introduction

The ageing process results in declines in muscle and neuromuscular function after the age of 30 [1,2]. This combined with increasing sedentary behaviour increases health complications and the risk of falling. It has been estimated that adults spend 51–68% of their waking time in sedentary behaviours [3], defined as activities done while sitting or reclining with energy expenditure ≤1.5 METs [4]. Indeed, technological and social factors have made sitting the most ubiquitous behaviour during working, domestic and recreational time [5,6]. Furthermore, both physical activity levels and the desire to be physically active decrease with advancing age, such that only 50% of all adults and only 25% of adults aged over 60 meet minimum physical activity guidelines [7]. Evidence from around the world also suggests that time spent in sedentary behaviours increased by ~20% as a result of COVID-19 lockdowns [8,9]. Sedentary behaviour, and a consequential lack of muscle contractile activity, drives numerous maladaptive metabolic and cardiovascular responses [10,11]. Worryingly, time spent in sedentary behaviours is associated with increased risk of at least 35 pathological and clinical conditions [12] and all-cause mortality [13], even after accounting for the effects of participating in recommended amounts of moderate and vigorous physical activity [14,15].

Sedentary behaviour also has a profound effect on neuromuscular physiology and, consequently, function. Experimentally-induced periods of muscle disuse, brought about by short-term limb immobilisation (a more extreme form of sedentary behaviour), have been demonstrated to decrease muscle cross-sectional area [16], decrease the ability to voluntarily activate motor units [17] and decrease motor unit firing rates [18,19]. These changes likely mediate the observed negative changes in neuromuscular output and ability to perform activities of daily living (ADLs) seen with sedentary behaviour. Observational studies have demonstrated that greater time spent in sedentary behaviour is negatively associated with muscle strength [20,21], whilst limb immobilisation studies have demonstrated decreased muscle force control during submaximal isometric contractions, assessed as an increase in the magnitude of force fluctuations (i.e., coefficient of variation) [22]. Importantly, maximal strength [23,24] and the ability to control force [25,26] are significant predictors of balance and other ADLs [27]. It is no surprise, therefore, that both observational [28,29] and experimental [30] studies have demonstrated decreases in the ability to maintain balance with increased sedentary behaviour.

Fortunately, it has been suggested that brief, frequent muscular contractions throughout the day can have a potent influence on key physiological processes that mediate the adverse effects of prolonged sedentariness [3,10]. Indeed, accumulating evidence suggests that interrupting long periods of sedentary time with regular physical activity “breaks” throughout the day has numerous health benefits, including increasing energy expenditure [31], protecting cardiovascular endothelial function [32], lowering blood pressure [33], improving postprandial glucose metabolism [34] and decreasing lower back musculoskeletal discomfort [35]. Furthermore, breaking up sedentary time is associated with decreased all-cause mortality [36]. These results suggest that relatively small changes in activity level and pattern have the potential to modify the adverse health risk of sedentary behaviour [6]. As such, minimising the amount of time spent in prolonged sitting and breaking up long periods of sitting as often as possible have now been included in governmental physical activity guidelines [37,38,39]. Nonetheless, the effects of breaking up sedentary time with physical activity breaks remains an understudied area, particularly in the context of neuromuscular function. Indeed, no study to date has investigated the effect of physical activity breaks on muscle strength, force control or balance.

Many studies on breaking up sedentary time have used either free walking or treadmill walking as their intervention [32,33,34,40]. However, these are not always viable options due to space and/or cost constraints [41]. Calisthenic exercises, such as body weight squats and lunges, represent an ideal alternative, in that they can be performed anywhere and without any equipment [31]. Previous research has demonstrated that breaking up short periods of sedentary behaviour (up to ~90 min) with calisthenics has a positive effect on energy expenditure and endothelial function [31,42]. A further advantage of calisthenics is that they target fitness components such as muscle strength and balance, which are key components of physical activity guidelines [37] and are specifically affected by sedentary behaviour [20,28]. To our knowledge, the use of calisthenics as short-term intervention to break up sedentary time during the working day has not been studied.

The aim of the present study was to investigate the effect of breaking up sedentary time with regular bouts of calisthenic exercise on neuromuscular function. To that end, we compared the effects of a four-week calisthenics intervention with a control group who carried on with their normal behaviour. Neuromuscular function was assessed as muscle strength (maximal voluntary contraction of the knee extensors), muscle force control (measures of the magnitude and complexity of knee extensor force fluctuations) and dynamic balance (the Y balance test).

## 2. Materials and Methods

### 2.1. Participants

Healthy adults aged 18–65 who spent ≥6 h a day sitting were recruited for the study (33 ± 13.1 years). Sitting time was initially assessed via self-report, although this was subsequently confirmed using accelerometery (activPAL4 micro; PAL Technologies Ltd., Glasgow, UK). Despite meeting the requisite daily sitting time, all participants were recreationally active but did not engage in any form of structured endurance or resistance training, nor did they train specifically for any sport. A total of 17 participants provided written informed consent to participate in the study, which was approved by the ethics committee of the University of Essex (Ref. ETH2021-0934), and which adhered to the Declaration of Helsinki. Participants were randomly assigned into a control group (*n* = 9) or a calisthenics intervention group (*n* = 8; see Table 1 for participant physical characteristics).

### 2.2. Experimental Design

Participants visited the laboratory on two separate occasions, five weeks apart. They were instructed to arrive at the laboratory in a rested state (having performed no strenuous exercise in the preceding 24 h) and to have consumed neither food nor caffeinated beverages in the 3 h prior to arrival. The first visit comprised familiarisation with testing equipment/experimental procedures and baseline (week 0) testing. Following this, all participants were instructed to carry on with their normal behaviour for the next week, so that baseline measures of sitting time could be confirmed using accelerometery. After this initial week, the control group were instructed to carry on with their normal behaviour for the next four weeks (weeks 1–4), while the calisthenics intervention group were given instructions on using calisthenics to break up sedentary time during the working week (i.e., 09:00–17:00, Monday–Friday) for the next four weeks (weeks 1–4; see “Section 2.5. *Calisthenics intervention*” below for further details).

### 2.3. Accelerometry

During their first visit to the laboratory, participants were provided with an ActivPAL4 physical activity logger (PAL Technologies Ltd., Glasgow, UK) to wear for the duration of the study. The ActivPAL4 is able to distinguish the incline of the leg to determine periods of lying, sitting, standing, and stepping [43]. The ActivPAL4 unit was placed into a nitrile sleeve and then attached to participants left leg at the midline of the anterior thigh using a Tegaderm film dressing (3M Health Care, St Paul, MN, USA). The nitrile sleeve and Tegaderm dressing providing a waterproof barrier, allowing participants to continue wearing the unit when bathing.

### 2.4. Measures of Neuromuscular Function

On arrival at the laboratory, participants were assessed on the Y balance test, a measure of single leg dynamic balance [44]. Assessment of dynamic balance was chosen given the dynamic nature of many ADLs [45]. The Y balance test apparatus consists of an elevated central footplate (2.54 cm off the ground) and pipes, with reach indicator blocks, attached in the anterior, posteromedial, and posterolateral directions. Participants stood with one leg on the footplate, with the most distal aspect of their foot on a marked stating line. While maintaining single leg stance, participants then reached with their free leg in the anterior, posteromedial, and posterolateral directions [46]. As a significant learning effect has previously been demonstrated, participants performed six practice trials in each of the three reach directions with each leg as the stance leg [47].

Following this practice, participants rested for 10 min, before performing three further attempts on each leg in which reach distance was recorded. A standardised testing order was used: participants started with their left leg as the stance leg and reached in the anterior, then posterolateral, and finally posteromedial directions. After doing this three times, participants repeated this with the right leg as the stance leg. All testing was conducted barefoot, to eliminate any additional balance and stability from the shoes [48].

Following completion of the Y balance test, participants rested for 10 min. They were then seated in the chair of a Biodex System 4 isokinetic dynamometer (Biodex Medical Systems Inc., Shirley, NY, USA), initialised and calibrated according to the manufacturer’s instructions. Their right leg was attached to the lever arm of the dynamometer, with the seating position adjusted to ensure that the lateral epicondyle of the femur was in line with the axis of rotation of the lever arm. Participants sat with relative hip and knee angles of 85° and 90°, respectively, with full extension being 0°. The lower leg was securely attached to the lever arm above the malleoli with a padded Velcro strap, whilst further straps secured firmly across the waist and both shoulders prevented any extraneous movement and the use of the hip extensors during the isometric contractions. The isokinetic dynamometer was connected via a custom-built cable to a CED Micro 1401-4 (Cambridge Electronic Design, Cambridge, UK). Data were sampled at 1 kHz and collected in Spike2 (Version 10; Cambridge Electronic Design, Cambridge, UK).

To assess knee extensor muscle strength, participants performed a series of brief (3 s) isometric maximal voluntary contractions (MVCs), each of which were separated by 60-s rest. Participants were given a countdown, followed by strong verbal encouragement to maximise their effort. Ten minutes after the establishment of muscle strength, participants performed a series of targeted isometric knee extension contractions at 10, 20 and 40% MVC, respectively, to assess their ability to control submaximal force. These contraction intensities were chosen because they are typical of the demands of ADLs [49]. The targets were determined from the highest instantaneous force obtained during the baseline (week 0) MVCs. Participants performed three contractions at each intensity, with contractions held for 6 s and separated by 4 s rest [26,50]. The intensities were performed in a randomised order, with 2 min rest between each intensity. Participants were instructed to match their instantaneous force with a 1 mm thick target bar superimposed on a display ~1 m in front of them and were required to continue matching this target for as much of the 6 s contraction as possible.

### 2.5. Calisthenics Intervention

Participants in the calisthenics intervention group were provided with a calisthenics exercise programme to complete during weeks 1–4. The calisthenics programme consisted of five different exercises: squats, arm circles, calf raises, knees to opposite elbows, and lunges [31,42]. Prior to conducting the intervention, participants were provided with written and verbal instructions, and demonstrations of each exercise were given. They were then provided with the opportunity to practice any unfamiliar exercises.

Participants were instructed to perform the exercises during the working day, notionally 09:00–17:00, Monday–Friday, ideally using them to break up sedentary time every hour (i.e., perform sets at 09:00, 10:00, 11:00, etc.). In week 1, participants were instructed to perform 4 sets of the exercises per day. One set consisted of 8 repetitions of each exercise, at a rate of one repetition every three seconds. Thus, each set of exercises took 2 min to complete. In week 2, participants were instructed to increase to 6 sets of the exercises per day and in weeks 3 and 4, participants were instructed to increase to 8 sets of the exercises per day. The reason for the progression in the number of sets across the intervention was twofold: (1) in order to conform to the progression principle of exercise training [51]; (2) pilot testing indicated that participants experienced delayed onset muscle soreness when performing 8 sets in the first week of the intervention, due to unaccustomed exercise, and this led to decreased adherence. Participants were asked to track their adherence to the intervention using a pre-prepared sheet. Based on this, it was estimated that participants completed 82% of the required calisthenics sets.

### 2.6. Data Analysis

The accelerometery data was analysed using the PAL software suite (Version 8; PAL Technologies Ltd., Glasgow, UK). Participants’ daily time spent sitting and daily step count were extracted and averaged across a five-day period (Monday–Friday; Table 1). Technical issues meant that baseline data were only available for 7 participants in the control group and 6 participants in the intervention group.

For the Y balance test, the greatest of the three trials was used for analysis of reach distance in each direction. Reach distance is significantly correlated with leg length [52]. As such, reach distance was normalised to leg length, measured as the distance in centimetres from the anterior superior iliac spine to the centre of the ipsilateral medial malleolus. The normalised value was calculated as: (reach distance/leg length) × 100. The normalised reach distance was, therefore, expressed as a percentage of leg length. The greatest normalised reach distance from each direction was also summed to yield a composite reach distance. Composite reach was calculated as: (sum of three reach directions/three times leg length) × 100.

Maximal muscle strength was determined as the highest instantaneous force obtained during the MVCs. For the submaximal force control tasks, the mean value of the three contractions at each intensity was calculated. Values for individual contractions were calculated based on the steadiest 5 s of each contraction, with MATLAB (R2017a; The MathWorks, Massachusetts, USA) code identifying the 5 s of each contraction with the lowest standard deviation (SD). It has been recommended that both magnitude- and complexity-based measures should be used when characterising force control [53]. Magnitude-based measures of force control characterise force steadiness, while complexity-based measures reflect force adaptability; that is, the ability to modulate force output rapidly and accurately in response to task demands [54].

The magnitude of variability in force output was assessed using the SD and coefficient of variation (CV), which provide measures of the absolute amount of variability in an output and the amount of variability normalised to the mean of the output, respectively [55]. The complexity of force output was assessed using approximate entropy (ApEn) and detrended fluctuation analysis (DFA) α. ApEn was used to assess the regularity or randomness of force output [56], while DFA was used to estimate temporal fractal scaling [57]. The calculations of ApEn and DFA are detailed in Pethick et al. [58]. In brief, ApEn was calculated with template length, *m*, set at 2 and the tolerance for accepting matches, *r*, set at 10% of the SD of force output, and DFA was calculated across time scales (57 boxes ranging from 1250 to 4 data points).

### 2.7. Statistics

All data are presented as means ± SD. Results were deemed statistically significant when *p* < 0.05. All data were tested for normality using the Shapiro–Wilk test. Independent samples *t*-tests were used to compare the physical characteristics (age, height, body mass) and baseline sedentary behaviour (sitting time, step count) of the control and calisthenics intervention groups. Students’ paired samples *t*-tests were used to compare the values for muscle strength, muscle force control (SD, CV, ApEn and DFA at 10, 20 and 40% MVC) and dynamic balance (anterior, posteromedial, posterolateral, and composite reach in each leg) obtained at baseline and in week 4 in the control and calisthenics intervention groups. This approach was chosen because this was a preliminary study and we wanted to assess the effects of the intervention and control conditions separately, rather than compare them across the two conditions. Cohen’s *d* was calculated for baseline and week 4 values for each variable, with effect sizes interpreted as trivial <0.2, small 0.2 ≤ *d* < 0.5, medium 0.5 ≤ *d* < 0.8, and large ≥ 0.8.

## 3. Results

### 3.1. Participant Physical and Sedentary Behaviour Characteristics

The physical and sedentary behaviour characteristics of the control and intervention groups are presented in Table 1. There were no significant differences between the control and calisthenics intervention groups for age (*p* = 0.237), height (*p* = 0.521), body mass (*p* = 0.370), baseline daily sitting time (*p* = 0.209) or baseline daily step count (*p* = 0.870). All participants for whom ActivPAL data were available met the criteria of at least 6 h a day sitting time.

### 3.2. Sedentary Behaviour

There were no changes from week 0 to week 4 in either the calisthenics intervention group or control group for daily sitting time (calisthenics: *p* = 0.332, *d* = 0.90; control: *p* = 0.146, *d* = 0.58) or step count (calisthenics: *p* = 0.814, *d* = 0.13; control: *p* = 0.980, *d* = 0148; Table 1).

### 3.3. Muscle Strength

There was a significant increase in knee extensor MVC from week 0 to week 4 in the calisthenics intervention group (*p* = 0.036; *d* = 0.21) but not the control group (*p* = 0.710, *d* = 0.05; Figure 1; Table 2).

### 3.4. Muscle Force Control Strength

The knee extensor force control data from weeks 0 and 4 are presented in Table 2. There were no changes in any of the force control measures during contractions at 10% MVC from week 0 to week 4 in the calisthenics intervention group (SD: *p* = 0.571, *d* = 0.14; CV: *p* = 0.658, *d* = 0.13; ApEn: *p* = 0.750, *d* = 0.55; DFA α: *p* = 0.638, *d* = 0.18). Similarly, there were no changes in any of the force control measures during contractions at 20% MVC from week 0 to week 4 in the calisthenics intervention group (SD: *p* = 0.208, *d* = 0.21; CV: *p* = 0.147, *d* = 0.24; ApEn: *p* = 0.077, *d* = 0.49; DFA α: *p* = 0.625, *d* = 0.08). There were significant decreases in SD (*p* = 0.031, *d* = 0.99) and CV (*p* = 0.016, *d* = 1.30) during contractions at 40% MVC from week 0 to week 4 in the calisthenics intervention group (Figure 2; Table 2). There were no changes in ApEn (*p* = 0.283, *d* = 0.44) or DFA α (*p* = 0.176, *d* = 0.70) during contractions at 40% MVC in the calisthenics intervention group.

There were no changes in any of the force control measures during contractions at 10% MVC (SD: *p* = 0.816, *d* = 0.06; CV: *p* = 0.786, *d* = 0.11; ApEn: *p* = 0.206, *d* = 0.57; DFA α: *p* = 0.133, *d* = 0.77), 20% MVC (SD: *p* = 0.442, *d* = 0.38; CV: *p* = 0.451, *d* = 0.23; ApEn: *p* = 0.631, *d* = 0.20; DFA α: *p* = 0.156, *d* = 0.99) or 40% MVC (SD: *p* = 0.303, *d* = 0.24; CV: *p* = 0.740, *d* = 0.15; ApEn: *p* = 0.286, *d* = 0.13; DFA α: *p* = 0.094, *d* = 1.00) from week 0 to week 4 in the control group.

### 3.5. Dynamic Balance

The Y balance test data from weeks 0 and 4 are presented in Table 2. In the calisthenics intervention group there was a significant increase in Y balance test normalised posterolateral reach distance with left leg stance from week 0 to week 4 (*p* = 0.046, *d* = 0.91; Figure 3). There were no changes in normalised anterior (*p* = 0.447, *d* = 0.19), posteromedial (*p* = 0.210, *d* = 0.46) or composite (*p* = 0.069, *d* = 0.50) reach distance in the calisthenics intervention group. There were no changes in normalised reach distance in any direction with the right leg from week 0 to week 4 in the calisthenics intervention group (anterior: *p* = 0.092, *d* = 0.63; posteromedial: *p* = 0.631; 0.20; posterolateral: *p* = 0.479, *d* = 0.26; composite: *p* = 0.159, *d* = 0.31). 

In the control group, there were no changes in normalised reach distance in any direction with either the left (anterior: *p* = 0.687, *d* = 0.11; posteromedial: *p* = 0.329, *d* = 0.25; posterolateral: *p* = 0.532, *d* = 0.21; composite: *p* = 0.450, *d* = 0.17) or right (anterior: *p* = 0.988, *d* = 0.00; posteromedial: *p* = 0.799, *d* = 0.06; posterolateral: *p* = 0.682, *d* = 0.04; composite: *p* = 0.780, *d* = 0.04) legs from week 0 to week 4.

## 4. Discussion

The major novel finding of the present study was that a short term (4-week) intervention designed to break up sedentary time with calisthenics exercise was effective at improving neuromuscular function. Specifically, the calisthenics intervention increased knee extensor MVC, knee extensor force steadiness (SD and CV) during contractions at 40% MVC and posterolateral reach with left leg stance in the Y balance test, whereas no such changes were evident in the control group. These results indicate the potential of calisthenics, i.e., simple body weight exercises that can be performed anywhere and without any equipment, as a tool to break up sedentary time and mitigate against its deleterious functional consequences.

### 4.1. Effect of Calisthenics on Neuromuscular Function

The present study adds to the growing body of literature demonstrating the positive physiological effects of interventions designed to break up sedentary time with physical activity [59,60]. It is the first experimental study to demonstrate that regularly breaking up sedentary time with calisthenics has a positive effect on objective measures of neuromuscular function (i.e., muscle strength, muscle force control and balance; Figure 1, Figure 2 and Figure 3; Table 2). This is in line with previous cross-sectional studies demonstrating better performance of ADLs with more frequent breaks in sedentary time [61]. Moreover, the presently observed improvements were evident with only 16 min per day of calisthenics exercise, spread across a period of 8 h; thus, providing evidence that small changes in activity level and pattern are able to modify neuromuscular function. Indeed, the daily requirement of the calisthenics intervention was such that it altered neither participants daily sitting time nor step count (Table 2). These results illustrate the utility of calisthenics as an intervention to break up sedentary time and improve neuromuscular function and serve to highlight their potential to improve other aspects of physiological function. Calisthenics exercises (depending on the exact ones chosen) can activate both upper and lower limb musculature, and can target multiple components of fitness [31,42], potentially making them more effective than previously used interventions such as free walking, treadmill walking or portable pedal machines. They can also be performed anywhere, including in the workplace or at home [31], and do not require any costly or space consuming equipment [41].

The results of the present study demonstrated that using calisthenics to break up sedentary time throughout the day can improve knee extensor strength (Figure 1; Table 2). The effectiveness of the calisthenics intervention was likely due to the squat and lunge exercises, both of which target the knee extensors [62,63]. Though no such measures were taken in the present study, previous research found no change in muscle thickness—indicative of hypertrophy—following 4 weeks of upper body calisthenics training [64]. As such, it is reasonable to suggest that the presently observed increase in strength was not due to muscle hypertrophy. Rather, it was likely mediated by decreases in motor unit recruitment threshold and increases in motor unit discharge rate [65]. Notably, decreased motor unit firing rates have been observed with enforced periods of inactivity [18,19]. The observed increase in strength has important health and functional implications. Lower extremity strength has been demonstrated to be predictive of all-cause mortality [66]. Indeed, for every 15 N increase in strength (similar to that observed in the present study; Table 2), there is a 7% reduced risk of all-cause mortality [67]. Knee extensor strength is also an important predictor of performance in ADLs, including balance [24] and locomotion [68].

The calisthenics intervention resulted in a significant decrease in the magnitude of knee extensor force fluctuations, as measured by the SD and CV, during contractions at 40% MVC (Figure 2; Table 2). These changes are indicative of increased force steadiness [54]. In contrast, the intervention had no effect on the complexity of force output, as measured by ApEn and DFA α, indicating no change in force adaptability [54]. The increase in force steadiness at 40% but not 10 or 20% MVC could be due to the use of gross motor movements (i.e., squats and lunges) that produce large overall forces [69]. Knee extensor EMG activity during the squat, for example, has been demonstrated to reach up to 63% of the value obtained during an isometric MVC [70]. Muscle force CV is strongly associated with variance in common synaptic input to motor neurons [27]. Thus, as with the increase in strength, it is likely that the increase in force steadiness was mediated by an increase in motor neuron output, specifically an increase in motor unit discharge rate [65]. The intervention-induced increase in knee extensor force steadiness has the potential to be as—if not more—important than the increase in strength. Indeed, previous research has demonstrated that force control is a stronger predictor of both static [25] and dynamic balance [26] than maximal strength.

Interestingly, Y balance test normalised posterolateral reach with left leg stance, but not anterior or posteromedial reach, increased as a result of the calisthenics intervention (Figure 3; Table 2). This increase in posterolateral reach is indicative of greater dynamic balance and stability [71]. The calisthenics exercises were, for the most part, linear exercises performed in the sagittal plane. Consequently, anterior reach, also performed in the sagittal plane, would have been expected to be the most likely of the reach directions to improve. Why only posterolateral reach increased as a result of the intervention is unclear, though it must be noted that previous training studies have often failed to demonstrate increases in all reach directions [72,73]. No improvements in reach with right leg stance were observed as a result of the intervention (Table 2). With the exception of anterior reach, the right leg was the better performing leg at week 0 (Table 2). Thus, the improvement in the left leg, but not the right leg, could simply be reflective of the left leg having greater scope to improve.

### 4.2. Limitations and Suggestions for Future Research

The present study was subject to several limitations, which can be broadly categorised as relating to the intervention, the sample size and population tested, and the study design. With regard to the intervention, it consisted of squats, arm circles, calf raises, knees to opposite elbows, and lunges, as previously used by Carter et al. [31]. Given the specificity principle of training, it could be argued that arm circles are unlikely to contribute to increases in knee extensor maximal strength, force control or dynamic balance. As such, future studies using calisthenics to break up sedentary time and improve neuromuscular function should ensure that all exercises used are specific to the outcomes being measured. Future studies could also manipulate the intensity (number of sets and repetitions) and/or duration of the intervention, thus conforming to the progression principle of training. Higher intensity or longer duration studies may be necessary to see improvements in the variables that did not change in the present study.

Our sample size was only 17 participants, with 8 of these undertaking the calisthenics intervention and the remaining 9 acting as the control group. The reason for this small sample size was that this was only a preliminary study. It should be noted, though, that this sample size was similar to that of many of the acute and chronic interventions for breaking up sedentary time highlighted by the systemic review of Benatti & Ried-Larsen [59]. With regard to the population tested, our participants comprised a mix of university students and staff (age range 18–59), who (ActivPAL data indicated) exhibited markedly different daily patterns of activity. It may be of utility to investigate more homogeneous populations, such as office workers, particularly given that sedentary behaviour can account for >80% of work time [74]. Indeed, given its simplicity to conduct, calisthenics has been suggested to be an ideal workplace intervention [42]. It may also be of interest to investigate an older adult (age ≥ 65 years) population, as both physical activity levels and desire to be physically active [7] both decrease with advancing age. Moreover, muscle maximal strength [75], the ability to control force [55,76] and the ability to maintain balance [77] all decrease with advancing age. The presently used calisthenics intervention would be easy for older adults to undertake in the comfort of their own homes and requires neither supervision nor equipment, all of which may help to increase adherence. 

Because the purpose of the study was simply to ascertain the effectiveness of breaking up sedentary time with calisthenics exercise on neuromuscular function, we did not take any measures that could determine the mechanism(s) for the intervention-induced improvements. Future studies could use techniques such as high-density electromyography to assess neural effects (e.g., motor unit recruitment and discharge properties) and imaging modalities such dual-energy x-ray absorptiometry to assess muscular effects (e.g., change in body composition/body size). Furthermore, we did not follow up with participants again beyond the end of the intervention. As such, it is unclear for how long the intervention-induced improvements in strength, force control and dynamic balance were retained.

## 5. Conclusions

Ageing and sedentary behaviour are associated with decreased maximal strength, decreased ability to control force and decreased ability to maintain balance. The present findings are the first to demonstrate that breaking up sedentary time with calisthenics exercise has a positive effect on these important indices of neuromuscular function. Calisthenics exercise overcomes the limitations of other physical activity interventions to break up sedentary time, in that it can be performed anywhere, without any equipment and targets multiple fitness components. Importantly, these results indicate that small changes in activity level and pattern (only 16 min per day, spread across 8 h) can mitigate against the negative effects of prolonged sedentary time. This may be a useful intervention to reduce the effects of ageing on neuromuscular function, potentially reducing future health complications.

## Figures and Tables

**Figure 1 ijerph-19-14597-f001:**
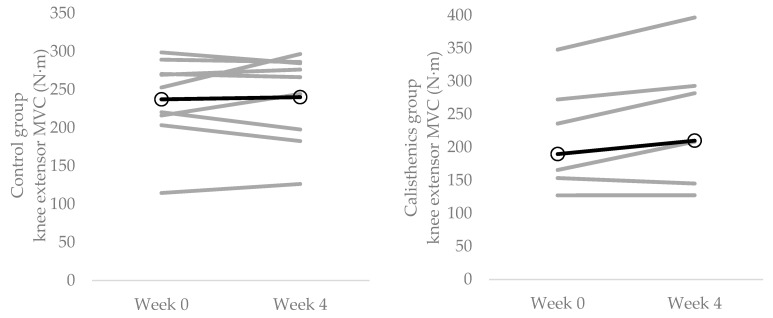
Individual changes (grey lines) in knee extensor MVC strength from baseline (week 0) to the end of the intervention (week 4). **Left panel** = control group; **right panel** = calisthenics intervention group. Dark lines represent group means.

**Figure 2 ijerph-19-14597-f002:**
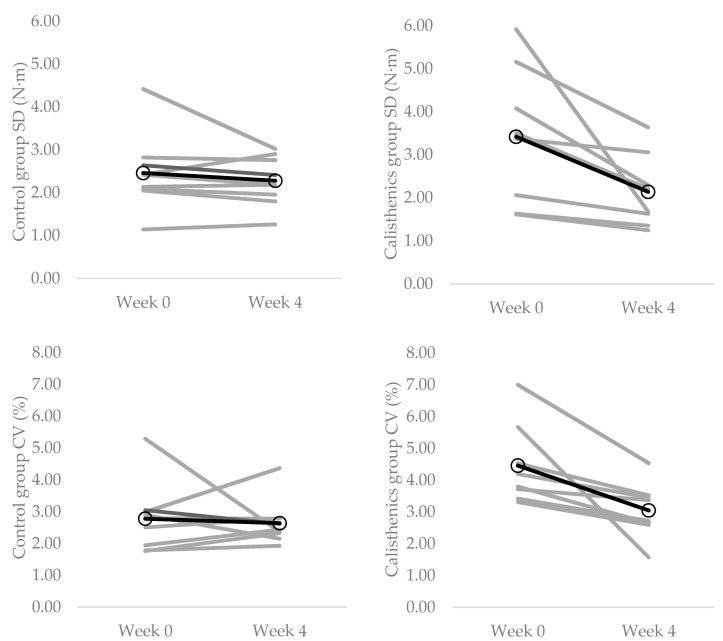
Individual changes (grey lines) in knee extensor force control during contractions at 40% MVC from baseline (week 0) to the end of the intervention (week 4). **Top left panel** = control group SD; **Top right panel** = calisthenics intervention group SD; **Bottom left panel** = control group CV; **Bottom right panel** = calisthenics intervention group CV. Dark lines represent group means.

**Figure 3 ijerph-19-14597-f003:**
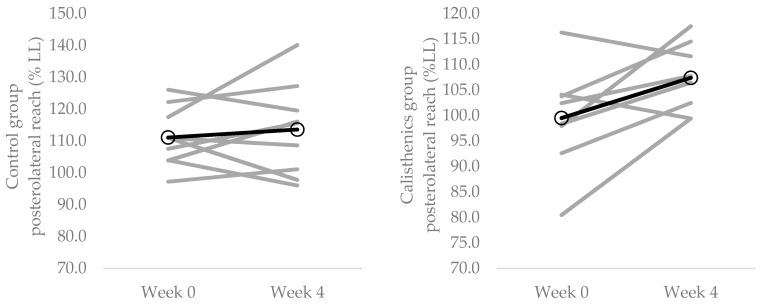
Individual changes (grey lines) in Y balance test posterolateral reach with left leg stance from baseline (week 0) to the end of the intervention (week 4). **Left panel** = control group; **Right panel** = calisthenics intervention group CV. Dark lines represent group means.

**Table 1 ijerph-19-14597-t001:** Participant physical and sedentary behaviour characteristics. Values are means ± SD.

	Control (*n* = 9)	Intervention (*n* = 8)
Age (years)	29.9 ± 9.7	37.4 ± 15.8
Height (m)	1.74 ± 0.07	1.71 ± 0.12
Body mass (kg)	81.7 ± 22.7	71.6 ± 22.6
Daily sitting time (hours) week 0	10.7 ± 1.7	9.1 ± 2.7
Daily sitting time (hours) week 4	9.3 ± 1.4	10.3 ± 1.1
Daily steps week 0	8354 ± 4919	7948 ± 3591
Daily steps week 4	7759 ± 4068	8456 ± 3720

**Table 2 ijerph-19-14597-t002:** Knee extensor strength, force control and Y balance test data from baseline (week 0) and the end of the intervention (week 4). Values are means ± SD.

	Control		Intervention	
	Week 0	Week 4	Week 0	Week 4
MVC (N·m)	237.4 ± 53.7	240.3 ± 58.5	189.8 ± 87.6	**210.2 ± 104.6**
SD				
10% MVC (N·m)	0.87 ± 0.20	0.86 ± 0.13	0.83 ± 0.33	0.79 ± 0.24
20% MVC (N·m)	1.24 ± 0.19	1.16 ± 0.23	1.37 ± 0.47	1.27 ± 0.85
40% MVC (N·m)	2.46 ± 0.88	2.28 ± 0.57	3.41 ± 1.60	**2.14 ± 0.85**
CV				
10% MVC (%)	3.81 ± 0.69	3.71 ± 1.03	4.29 ± 0.99	4.17 ± 0.81
20% MVC (%)	2.97 ± 0.97	2.78 ± 0.64	3.74 ± 1.00	3.49 ± 1.09
40% MVC (%)	2.78 ± 107	2.64 ± 0.71	4.46 ± 1.27	**3.05 ± 0.87**
ApEn				
10% MVC	0.83 ± 0.12	0.78 ± 0.03	0.85 ± 0.10	0.80 ± 0.08
20% MVC	0.73 ± 0.13	0.71 ± 0.05	0.76 ± 0.14	0.70 ± 0.10
40% MVC	0.59 ± 0.09	0.60 ± 0.05	0.52 ± 0.17	0.58 ± 0.09
DFA α				
10% MVC	1.04 ± 0.07	0.99 ± 0.06	1.00 ± 0.10	1.03 ± 0.21
20% MVC	1.16 ± 0.07	1.10 ± 0.05	1.14 ± 0.11	1.13 ± 0.14
40% MVC	1.30 ± 0.05	1.25 ± 0.05	1.29 ± 0.09	1.23 ± 0.08
Y balance test (left stance)				
Anterior reach (% LL)	65.4 ± 10.0	64.8 ± 8.0	62.1 ± 6.3	63.3 ± 6.1
Posteromedial reach (% LL)	111.5 ± 14.3	115.4 ± 16.3	102.9 ± 10.3	106.9 ± 6.9
Posterolateral reach (% LL)	111.1 ± 8.9	113.6 ± 14.5	99.5 ± 10.3	107.4 ± 6.8
Composite reach (% LL)	104.2 ± 11.3	106.4 ± 14.3	97.7 ± 11.6	102.4 ± 6.5
Y balance test (right stance)				
Anterior reach (% LL)	64.5 ± 6.5	64.5 ± 6.3	60.9 ± 5.6	64.2 ± 4.8
Posteromedial reach (% LL)	113.7 ± 14.1	112.8 ± 14.6	106.4 ± 7.7	107.8 ± 6.3
Posterolateral reach (% LL)	112.0 ± 12.6	111.4 ± 14.9	104.0 ± 5.9	106.0 ± 9.2
Composite reach (% LL)	105.2 ± 12.9	104.7 ± 12.9	100.4 ± 8.5	102.8 ± 7.0

MVC = maximal voluntary contraction; SD = standard deviation; CV = coefficient of variation; ApEn = approximate entropy; DFA = detrended fluctuation analysis; LL = leg length. **Bold** indicates a significant difference (*p* < 0.05) from Week 0.

## Data Availability

Supporting data are available from J.P. (jp20193@essex.ac.uk) on request.

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
