# Peer review of "The Effect of Breaking Up Sedentary Time with Calisthenics on Neuromuscular Function: A Preliminary Study"

_ijerph, 2022, doi:10.3390/ijerph192114597_

Round 1

Reviewer 1 Report

Introduction

No comments. Well done.

Methods

calisthenics intervention: Can you please clarify how the sets were distributed? Were participants instructed to specifically perform them X hours apart, etc?

calisthenics intervention: Did you ask them to track their adherence to the intervention?

Results

Figures: could you please put titles for each graph in each sub-figure instead of in the main figure title (e.g. instead of Left panel = control group write control group as an additional x-axis label); this would be especially helpful in figure 2

Figure 2: which test are these data from? 10%, 20%, or 40% MVC? From paragraph text at line 270, seems to clearly indicate 40% MVC, but would be good to have in figure title as well.

Discussion

This isn't an actionable suggestion, but I just wanted to say that I think this sentence in the limitations about using arm circles: "As such, future studies using calisthenics to break up sedentary and improve neuromuscular function should ensure that all exercises used are specific to the outcomes being measured;" is being a bit harsh on yourselves.

Conlusions

Clear and supported

Author Response

Reviewer 1 comments: Methods calisthenics intervention: Can you please clarify how the sets were distributed? Were participants instructed to specifically perform them X hours apart, etc?

Response: this has been added Line 182 Participants were instructed to perform the exercises during the working day, notionally 09:00-17:00, Monday-Friday, ideally using them to break up sedentary time every hour (i.e., perform sets at 09:00, 10:00, 11:00, etc.)

reviewer 1 comment: Methods calisthenics intervention: Did you ask them to track their adherence to the intervention?

Response: Line 194: Participants were asked to track their adherence to the intervention using a pre-prepared sheet. Based on this, it was estimated that participants completed 82% of the required calisthenics sets.

Results

Reviewer 1 comments Figures: could you please put titles for each graph in each sub-figure instead of in the main figure title (e.g. instead of Left panel = control group write control group as an additional x-axis label); this would be especially helpful in figure 2

Response: this has been actioned.

Reviewer 1 comment: Figure 2: which test are these data from? 10%, 20%, or 40% MVC? From paragraph text at line 270, seems to clearly indicate 40% MVC, but would be good to have in figure title as well.

Response: this has been actioned

Reviewer 2 Report

Review ijerph 6853: The effect of breaking up sedentary time …….

This research is interesting and useful. The issue of sedentary lifestyle is important, affects all ages, and has not been resolved. The aim of the present study was to investigate the effect of breaking up sedentary time with regular bouts of calisthenic exercise on neuromuscular function. The problem is that since the group of participants is very heterogeneous and not very numerous, other variables that were not considered may have intervened. For this reason, in our perception, more details about the profile of the participants would have been convenient. For example, it is not evident if the sample contained people who are usually athletes together with others with a marked history of sedentary lifestyle. The characteristics data (Table 1) refer only to the 4 weeks of the investigation. We know that the participants are a heterogeneous sample, in the age range 18-59, and in different daily patterns of activity, as the authors express in the limitations of the study

In either case, the frame of reference is focused and up to date on the topic. The methodology is carefully detailed and seem sufficient. The results are reliable, and consistent. It can be said that it meets the standards. Despite being very specific, it meets enough quality. 

A small observation, however, the authors should not claim that “Calisthenics exercises can activate both upper and lower limb musculature, and target multiple components of fitness, making them more effective than previously used interventions such as free walking, treadmill walking or portable pedal machines” since this comparison does not emerge from the present investigation. This research is very specific and punctual, then the discussion must not exceed its limits.  

After these observations, and in summary, we can support the publication of the research with the certainty of its interest and its contribution to an urgent topic.

Author Response

Reviewer 2 comment: The problem is that since the group of participants is very heterogeneous and not very numerous, other variables that were not considered may have intervened. For this reason, in our perception, more details about the profile of the participants would have been convenient. For example, it is not evident if the sample contained people who are usually athletes together with others with a marked history of sedentary lifestyle.

Response: this has been addressed in line 102 Despite meeting the requisite daily sitting time, all participants were recreationally active but did not engage in any form of structured endurance or resistance training, nor did they train specifically for any sport.

Reviewer 2 comment: the authors should not claim that “Calisthenics exercises can activate both upper and lower limb musculature, and target multiple components of fitness, making them more effective than previously used interventions such as free walking, treadmill walking or portable pedal machines”

Response: this has been removed.